# Factors influencing the adoption and participation rate of nursing homes staff in a saliva testing screening programme for COVID-19

Benoit Pétré [1]*, Marine Paridans[1], Nicolas Gillain[1], Eddy Husson[1], Anne-Françoise Donneau[1,2,3], Nadia Dardenne[1,3], Christophe Breuer[4], Fabienne Michel[2,5], Margaux Dandoy[6], Fabrice Bureau[6,7], Laurent Gillet [6,8,9], Dieudonné Leclercq[1], Michèle Guillaume[1,2]

1 Department of Public Health, University of Liege, Liège, Belgium, 2 Risk Assessment Group COVID-19, Liège University, Liège, Belgium, 3 University and Hospital Biostatistics Center (B-STAT), Faculty of Medicine, University of Liège, Liège, Belgium, 4 Liège University, Liège, Belgium, 5 Collection and Analysis of Data and Information of Strategic Utility (RADIUS), Liège University, Liège, Belgium, 6 Covid-19 Platform, Liège University, Liège, Belgium, 7 Laboratory of Cellular and Molecular Immunology, GIGA Institute, Liège University, Liège, Belgium, 8 Fundamental and Applied Research for Animal and Health (FARAH) Center, Liège University, Liège, Belgium, 9 Laboratory of Immunology-Vaccinology, Liège University, Liège, Belgium

☯ These authors contributed equally to this work.

* benoit.petre@uliege.be

**Data Availability Statement:** All relevant data are within the paper and its Supporting Information files.

## Abstract

Testing strategies are crucial to prevent and control the spread of covid-19 but suffer from a lack of investment in understanding the human factors that influence their implementation. The aim of this study was to understand the factors that encourage participation and the level of engagement of nursing homes staff in a routine saliva testing programme for COVID-19 In December 2020, nursing homes (n = 571) in Wallonia (Belgium) were invited to participate in a saliva testing programme for their staff. The directors were questioned by telephone at the end of a 3-week pilot phase. 445 nursing homes took part in the evaluation questionnaire, of which 36(8%) answered that they chose not to participate in the testing programme. The average participation rate of nursing staff was 49(±25)%. Perception of the justification of the efforts required for testing and perception of practicability of the procedure were significantly associated with the adoption of the system by the nursing homes directors (OR(95%CI): 5.96(1.97–18.0), p = 0.0016); OR(95%CI): 5.64(1.94–16.4), p = 0.0015 respectively). Staff support, incentives and meetings increased the level of engagement in testing (p<0.05). While the adoption of the programme confirmed the acceptability of salivary testing as a means of screening, the participation rate confirmed the need for studies to understand the factors that encourage health care staff to take part. The results suggested rethinking strategies to consider staff engagement from a health promotion perspective.

**Funding:** The authors received no specific funding for this work.

**Competing interests:** FB and LG are the inventors of the device used in the saliva collection kit. This device was patented (WO2022013235A1) and produced by Diagenode (Seraing, Belgium) under a commercial agreement with the University of Liège. The authors report no other conflicts of interest in this work.

## Introduction

For many months, the coronavirus (COVID-19) pandemic has challenged the organisation and effectiveness of our health care system, particularly in the areas of prevention and health promotion. Prevention is essential both for individuals and community in order to prevent the spread of the virus, avoid saturation of health care services and also to combat the psycho-medico-social consequences of the crisis. Therefore, in addition to individual prevention measures, early detection, triage and effective isolation of potentially infected and infectious patients are essential to prevent unnecessary community exposure [1] and to break the chains of transmission.

In Europe, elderly people living in nursing homes have been severely affected. While the population of nursing homes represents less than 1% of the total population in European countries, their residents accounted for 31–80% of all deaths in the first wave [2, 3]. Because of their age, multimorbidity, immunodeficiency, frailty, communal living and also the fact that they represent an open community (through the entries / exists of health care staff or visitors who may carry the virus), residents of nursing homes have been the most heavily impacted by the SARS-CoV-2 pandemic and constitute a "vulnerable" category of people to be protected. The European Centre for Disease Prevention and Control (ECDC) published a series of recommendations [4], in May 2020, to try to limit morbidity and mortality in nursing homes. Systematic screening of staff was highlighted as a priority as staff represent one of the main causes of the increased spread of COVID-19 in healthcare facilities [4]. This is especially true when visits of residents are suspended or strictly limited [5, 6].

The implementation of such systematic testing depends on the capacity of governments to organise and to promote its use among beneficiaries. While the first approach concerns questions of governance [7], this paper focuses on questions related to the participation of beneficiaries in a COVID-19 testing scheme that was offered to them. The panic that emerged from the pandemic reactivated paternalistic and biomedical approach of governance that put health promotion on the back burner over the crisis (mainly at the beginning) [8, 9].

In particular, the issue of adherence to a Walloon region proposal for systematic and gracious screening (by saliva sampling) for COVID-19 by staff working in nursing homes was considered. A large volume of literature, identifying barriers and incentives for individuals, was used as the basis for the evaluation of participation in preventative health practices. Among the best-known psycho-sociological theories were (A) the Health Belief Model [10], which focused on the beliefs (convictions) of vulnerability, threat and benefit that lead to the decision about whether to participate in a screening programme for diseases; (B) Bandura's Self Efficacy de [11], which focused on the self-assessment of one's capacity to act effectively; (C) Fishbein & Ajzen's [12] theory of reasoned action, which focused on personal values and social norms; and (D) Rotter's [13] locus of control, which focused on how causal attributions affect what happens to us (internal or within our control, or external, beyond our control). These models have been widely used and are indeed the subject of integrative proposals such as that of Leclercq [14]. In his integrative model "ASCID", Leclercq demonstrated that a decision to take preventive health measures and to stick with them is the result of a (problem-customized) combination of several factors influencing, for example, norms (real or perceived), knowledge, technical know-how and material availability. He unifies these models by articulating the various factors according to the logic of decision theory. Decision theory states that the decision-maker can choose between several actions (in this case adopting or not adopting saliva testing), that each action has positive (attractive) or negative (repulsive) consequences for the decision-maker, and that the decision-maker subjectively attributes a subjective value and a probability of occurrence to each consequence. The "balance of decisions" refers to the

probable benefits that lead to the final decision (the one minimising costs and maximising benefits). These decision theories have so far been scarcely applied to inspire the promotion of preventive measures against COVID-19.

The objective of this paper was to understand the factors linked to the adoption and degree of engagement of nursing homes staff in a systematic screening programme for COVID-19 by saliva testing.

## Materials and methods

### Context

Belgium has a high proportion of aging population with approximately 19% of the population older than 65 years. 5.7% of the latter group is living in NHs. Belgium has one of the highest number of NH beds per 1000 population in Europe (50 NH beds per 1000 population over 60) [15] and tops the chart with the number of people of 85 years and older living in an institution. Belgium was also ranked among the countries with the highest COVID-19 mortality in the world during its first COVID-19 wave [6, 16].

In view of the disastrous situation in Belgium with regard to COVID cases, the Walloon Government (that is responsible for the health of elderly, particularly the management of NHs) wished to help reduce the burden on the NHs and financially supported: the provision of saliva sampling kits; the analysis of samples by PCR; the delivery of results; and the logistical operations necessary for the implementation of the system. However, this evaluation study was not funded by the Walloon Government and was conducted completely independently by the authors of the paper. The results were communicated to the Government in order to adjust the strategy for implementing testing.

The nursing homes located in the south of Belgium (French-speaking part—Walloon region) (n = 571) were invited by the regional authority to participate in a free of charge saliva testing program in December 2020. Saliva testing is now a complementary means to nasopharyngeal testing in the routine testing strategy [17]. The test was offered to all employed staff, self-employed staff working in the institution, trainees and volunteers (administrative, health care and other worker staff estimated over 35.000 people) both for a 3-week pilot phase.

The saliva testing procedure (sensitivity of 68% [18]) was subject to a standard organizational procedure as previously described [19–21] and summarized in five steps in Box 1.

### Box 1. General organisation of saliva testing for the Walloon nursing homes.

1- Distribution of kits to staff to be tested

The distribution of the self-collection-of-saliva kits was carried out from 13 distribution points spread throughout the Walloon region. Each distribution point had its own premises and opening hours and each participating institution nominated one person to go to their collection point to obtain the self-collection kits. Within each nursing homes, the directors organised the collection of the kits from the collection point and distribution to the whole staff. The internal organisation of the distribution was the responsibility of the direction of the institution.

2- Self-Collection of the sample by the worker

Each worker took a saliva sample using the kit provided, following the instructions in the received individual box. The sample was taken and collected on the same day for all

the workers of a same institution. The date was communicated earlier to the workers by the nursing homes direction. Each worker recorded (noted or photographed) the barcode noted on their sample tube so that their individual result could be confidentially consulted by themselves afterwards.

3- Depositing the sample by each worker in the nursing home collection point.

The sample collection was organised so that all samples from the institution on the designated day could be assembled and carried to the distribution point. The nursing homes organised the collection of samples from staff members using specific equipment of proctection made available to them for a safe transport of the samples. It means that the data collection was at one point in time for EACH nursing home over the week. A specific time slot was dedicated for each NH. But the differents NH had different moments of saliva samples collection

4- Delivery of samples to the collection point.

The direction of the institution, or the person they nominated for this purpose, ensured the safe (from a hygiene point of view) delivery of the samples to the collection point. At closing time, all samples received were immediately transferred to the laboratory for analysis.

5—Communication of results.

As soon as an analysis was completed, the laboratory communicated the result via a secure web platform. The results could be obtained in two ways: the person who took the sample could obtain his/her individual result by entering the recorded barcode number of their sample (anonymity was respected, since only the participant knew their barcode), or the nursing homes direction could consult a statistical report of anonymized and aggregated results for their establishment.

## Study design

This was an evaluative, quantitative and cross-sectional study. The evaluation was carried out at the end of the 3-week pilot phase of the testing.

## Study population

This included all 571 nursing homes in the Walloon region. Persons who answered the survey had to be the director of the institution or a member of the management (deputy director or the head of nursing involved in the organisation of the testing) and, in addition, had to speak French.

## Study parameters and data collection tools

The data for this study came from two sources: (1) a home-made questionnaire administered to the management of the nursing homes via telephone interviews conducted by staff trained in a standardised approach; (2) the transmission by the AVIQ of the aggregated results of participation in saliva testing by nursing homes for each week of the pilot phase.

**Explanatory variables studied.** The explanatory variables were developed on the basis of Leclercq (2010)'s model and were estimated as follows in the questionnaire

- Characteristics of the institutions: sector (commercial, associative or public), geographical distribution (province), size (number of staff), location of the nursing homes in relation to the distribution points (estimated time to reach the distribution point)

- Importance given to saliva testing. The directors were asked to give marks to 4 indicators via a binary modality (Yes/No):

  ○ Added value of saliva testing against the spread of the virus compared to barrier and other screening methods

  ○ Influence of the sanitary situation in the nursing homes on the intensity of saliva testing efforts

  ○ Effectiveness of carrier detection by saliva testing

  ○ Early detection of virus carriers by saliva testing

- The impression of the procedure for implementing testing. The managers were asked to rate three indicators via a binary modality (Yes/No):

  ○ Legibility of the procedure

  ○ Accuracy of the procedure

  ○ Practicality in terms of the workload required by the procedure

- Difficulties encountered during the implementation of saliva testing. Directors were asked to indicate whether they had encountered any difficulty in one or more of the following steps by means of a binary modality (Yes/No):

  ○ Self-sampling

  ○ Test results

  ○ Schedule

  ○ Deposit of samples at the nursing homes

  ○ Transport of kits

  ○ Distribution of kits to staff

- The importance given to testing. This aspect was approached by 2 variables: the reception of testing in the nursing homes (Very positive—Somewhat positive—Somewhat negative/negative) and the priority given to testing (High vs Medium/Low)

- The way in which the decision to take part in testing was made (by direction, by the organising authority or jointly), the type of support provided by workers themselves (Yes/No/mitigated) and by the hierarchy (total/partial/none)

- How the workers were motivated to take part in testing:

  ○ Freedom to choose whether to participate or not (full, incentives, mandatory)

  ○ Presence or absence of incentives: appointment of a coordinator, e-mail notification, special meeting, video tutorial (Yes /No).

  **Dependent variables.**   Two dependent variables were studied:

- Adoption by the nursing home of the testing scheme: this binary variable indicates whether or not each nursing home took part in the proposed saliva testing programme.

- The level of staff engagement in the nursing home: this variable is approximated by the average weekly participation rate of each nursing homes staff during the screening period.

## Organisation of data collection

The study was carried out according to the following procedure:

1. E-mail notification: Directors or the equivalent were contacted by e-mail (by telephone if the e-mail address was incorrect) to notify them of the implementation of the survey.

2. Appointment scheduling: contact was made by telephone to answer any questions from the nursing homes and to arrange a telephone appointment with the director (or equivalent) who had agreed to their participation in the survey. If requested, the questionnaire was sent by e-mail. A maximum of three telephone reminders (call back) were made in cases of non-response in order to obtain as many participants as possible.

3. Telephone interviews were made by trained interviewers to ensure proficiency in data collection tool and standardisation in the interview. Debriefing meetings were held with the pool of interviewers at the end of each day of data collection.

4. Computer encoding: a computer interface was created to allow interviewers to encode the data collected during telephone contacting.

## Data analysis

For qualitative parameters, results were expressed as numbers and frequencies. For quantitative variables, they were expressed as means and standard deviations (SD), medians (P50) and interquartile ranges (Q1-Q3) and as ranges (Min-Max). The normality of the distribution of each quantitative parameters was verified using the mean-median comparison, by a histogram and Quantile-Quantile plot and tested with the Shapiro-Wilk hypothesis test.

For the binary outcome (adoption), association between the outcome and each independent variables was assessed using univariate binary logistic regression. A multivariate analysis was then carried out including all significant ($p < 0.05$) independent variables in univariate approach. Results were presented using odds ratio (OR) and corresponding 95% confident interval (CI95%).

For the quantitative outcome (level of staff engagement), the association between that outcome and quantitative independent parameters was assessed by means of a Pearson or Spearman correlation. Association with qualitative parameters, however, was assessed using the student's t-test for independent samples or a one-way analysis of variance (ANOVA-1) for independent parameters with more than two categories. When normality assumption was not fulfilled, the non-parametric Mann-Whitney test or Krukal-wallis test were considered. When required, multiple comparisons between groups were carried out. Finally a multiple regression was performed with all significant independent parameters. The adjusted R-squared ($R^2$) was also provided, as a quality index of the regression.

Results were significant at the 5% critical level ($p < 0.05$). The statistical analyses were carried out using SAS (version 9.4 for Windows) statistical package and RStudio.

## Ethical considerations

The ethics committee of the University Hospital of Liège (Comité d'Ethique Hospitalo-Universitaire de Liège, Belgium) determined this study was exempt from review and did not oppose the undertaking of the study and waived the recording of a written consent. An information

letter was sent to all potential participant. As the participation was based on a voluntary basis and as the participation to the survey was made by phone call, we collected only oral consent from NH directors.

The testing device has been designed to deliver the result of the samples in a completely anonymous way. The result of an analysis is delivered on presentation of a barcode number of the corresponding sample, without being associated with an identity. Consequently, only the person who has taken a self-sample of saliva and who has the corresponding barcode number can make the link between him or herself and the result delivered on the interface (website). Only statistical, aggregated information was communicated to the organisation that requested the tests. There is no personal data that are transferred.

Neither the Walloon region, nor the NH authorities, nor the University of Liège are able to link a sample to the person concerned; nor are they able to determine which persons have or have not taken part in regular tests. It was therefore not possible to request nominative consent. Nevertheless, when consulting the results via the computer interface, persons who had submitted a sample and wished to know the result of the analysis were informed about the uses of their salivary testing to guide the objectives of the current study.

## Results

### Study participants

From a total of 571 eligible nursing homes in the Walloon region, the final interviewed sample included 445 nursing homes (i.e. 77.9% of the eligible population) (Fig 1). Losses were mainly due to unsuccessful contacts with the nursing homes or decision to not participate to the survey. In 75% of the 445 cases the survey respondent was the directors of the nursing homes or equivalent management member (12% charge nurse and 4% management advisor, 9% other).

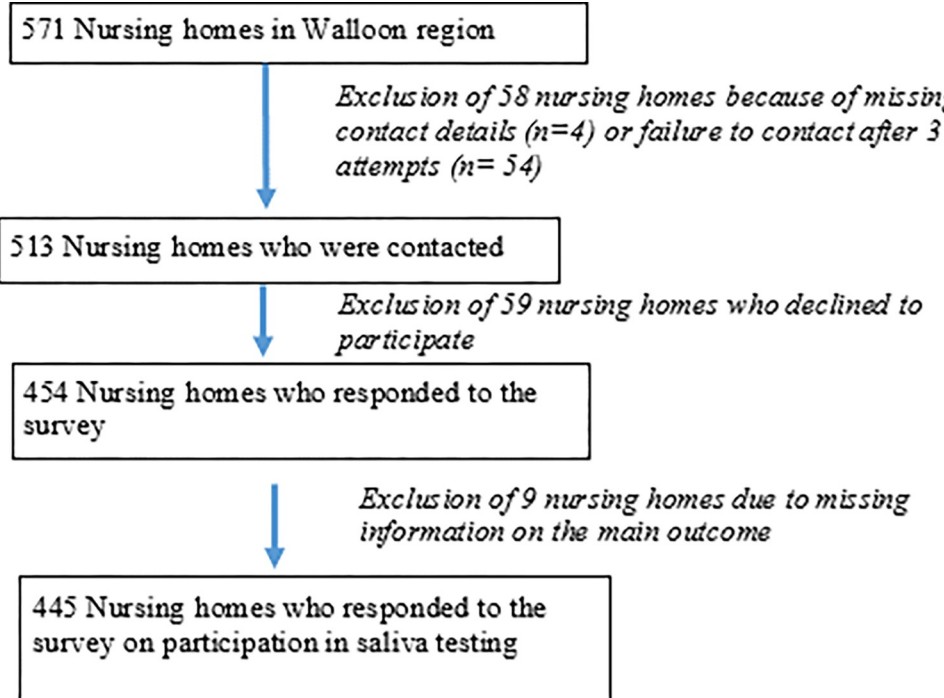

**Fig 1. Flowchart of participation in the survey.**

More than half (66.5%) of the 445 replies came from personnel who had worked in the institution for at least 3 years.

## Factors favouring the adoption of saliva testing

Table 1 describes the factors favouring adoption of saliva testing by direction of nursing homes. Of the 445 nursing homes that took part in the study, 36 (or 8%) chose not to participate in the testing scheme.

Among the different variables considered in the study with regard to the adoption of the saliva testing system, all the variables relating to the value placed on testing and the perception of the implementation of the procedure were significantly associated with the adoption of testing (p <0.05). The adoption of testing was more successful for nursing homes for whose directors had a positive opinion of the value of testing (justification of perceived efforts (p<0.0001), effective (p = 0.04), early detection of symptoms (p = 0.011), and perceived added value compared to other preventative measures (p<0.0001)), together with a positive view of the procedure to be implemented (legibility, precision and practicability of the procedure). The adoption of testing was also more successful for nursing homes in the public sector, compared to commercial sector (p = 0.034).

In the multivariate model, the perception about the "Justification of testing requirements" and the practicability of the procedure played a significant part in the adoption of the system (respective ORs(CI) of 5.96 (1.97–18.0) (p = 0.0016) and 5.64 (1.94–16.4) (p = 0.0015)).

## Factors enhancing the level of staff engagement in saliva testing

The average participation rate of nursing homes over the 3 weeks was 49%.

The factors that contributed to this level of participation are described in Table 2.

The level of engagement of nursing homes in the saliva screening system was significantly influenced by their geographical location within Wallonia (p<0.0001). This result was confirmed in the multivariate analysis. The province of Luxembourg stood out from the other four Walloon provinces with a higher than average participation rate among the staff of 59% for nursing homes located in this area.

In terms of the value attributed to testing, only the perceived added value of testing compared to other prevention measures significantly influenced staff participation in testing (51 ±25% of average engagement vs 38±24% for the nursing homes which were more negative about this testing system, p = 0.0011).

However, this result did not remain significant in the multivariate model (p = 0.215).

Factors relating to the perception of the procedure are not determinants significantly associated with the participation rate of nursing homes workers (p>0.05).

Perceived support from management and staff increased staff participation in testing (p = 0.022 and <0.0001 respectively) although in the multivariate model only staff support remained significant.

The perceived positive response of directors (Personal acceptance of testing by the direction) and the high priority given by direction to saliva testing were associated with higher levels of staff participation (p = 0.003 and p<0.0001 respectively). In the multivariate model, only high priority for testing remained significantly associated with higher staff participation (p = 0.002).

In terms of staff participation, encouraging or imposing testing on staff increased the likelihood of participation (55±23% and 62±26% vs 46±26% participation when total freedom of choice was given p = 0.0003). In the multivariate model, only the incentive remained more favourable than freedom of choice (p<0.001). The mobilisation of staff through the holding of

**Table 1. Factors determining the adoption of a systematic saliva testing system by Walloon nursing homes for staff to control the spread of Covid-19 (n = 445, December 2020).**

| Variable | Categories | N | All Number (%) | N | Yes Number (%) | N | no Number (%) | Univariate OR (IC95%) | p-value* | Multivariate OR (IC95%) | p-value* |
|---|---|---|---|---|---|---|---|---|---|---|---|
| *Characteristics of the Institution* | | | | | | | | | | | |
| Sector | | 445 | | 409 | | 36 | | | 0.10 | | |
| | COMMERCIAL | | 227 (51.0) | | 204 (89.9) | | 23 (10.1) | 0.31 (0.14–0.91) | **0.034** | / | / |
| | ASSOCIATIVE | | 99 (22.2) | | 90 (90.9) | | 9 (9.1) | 0.35 (0.10–1.17) | 0.087 | / | / |
| | PUBLIC | | 119 (26.7) | | 115 (96.7) | | 4 (3.3) | Ref | Ref | / | / |
| Geographic Distribution | | 445 | | 409 | | 36 | | | 0.94 | | |
| | LIEGE | | 133 (29.9) | | 121 (90.9) | | 12 (9.1) | 0.74 (0.20–2.75) | 0.65 | / | / |
| | NAMUR | | 66 (14.8) | | 60 (90.9) | | 6 (9.1) | 0.73 (0.17–3.09) | 0.67 | / | / |
| | HAINAUT | | 172 (38.7) | | 160 (93.0) | | 12 (7.0) | 0.98 (0.26–3.62) | 0.97 | / | / |
| | LUXEMBOURG | | 30 (6.7) | | 27 (90.0) | | 3 (10.0) | 0.66 (0.12–3.51) | 0.62 | / | / |
| | BW | | 44 (9.9) | | 41 (93.2) | | 3 (6.8) | Ref. | Ref | / | / |
| Distance from the collection centre | | 444 | | 408 | | 36 | | | 0.74 | | |
| | <11 | | 149 (33.6) | | 139 93.3) | | 10 (6.7) | 1.38 (0.58–3.25) | 0.46 | / | / |
| | 11–16 | | 151 (34.0) | | 138 (91.4) | | 13 (8.6) | 1.05 (0.47–2.36) | 0.90 | / | / |
| | >16 | | 144 (32.4) | | 131 (90.9) | | 13 (9.1) | Ref | Ref | | |
| Nursing homes Size | Number of personnel | 445 | 60 (40–87) | 409 | 60 (40–90) | | 54,5 (30,5–72) | 1.01 (0.99–1.02) | 0.070 | / | / |
| *Perceived value of testing* | | | | | | | | | | | |
| Justification of testing requirements | | 433 | | 403 | | 30 | | | | | |
| | Yes | | 365 (84.3) | | 349 (95.6) | | 16 (4.4) | 5.66 (2.61–12.2) | < **.0001** | 5.96 (1.97–18.0) | **0.0016** |
| | No | | 68 (15.7) | | 54 (79.4) | | 14 (20.6) | Ref | Ref | | |
| Screening effectiveness | | 408 | | 381 | | 27 | | | | | |
| | Yes | | 322 (78.9) | | 305 (94.7) | | 17 (5.3) | 2.36 (1.04–5.36) | **0.040** | 1.62 (0.30–8.78) | 0.57 |
| | No | | 86 (21.1) | | 76 (88.4) | | 10 (11.6) | Ref. | Ref | | |
| Early detection | | 394 | | 369 | | 25 | | | | | |
| | Yes | | 317 (80.5) | | 302 (95.3) | | 15 (4.7) | 3.01 (1.29–6.98) | **0.011** | 0.83 (0.15–4.53) | 0.83 |
| | No | | 77 (19.5) | | 67 (87.0) | | 10 (17.0) | Ref. | Ref | | |
| Perceived added Value Compared to Other prevention Measures | | 433 | | 402 | | 31 | | | | | |
| | Yes | | 376 (86.8) | | 358 (95.2) | | 18 (4.8) | 5.88 (2.70–12.8) | <**0.0001** | 2.31 (0.57–9.36) | 0.24 |
| | No | | 57 (13.2) | | 44 (77.2) | | 13 (22.8) | Ref. | Ref | | |
| *Perception of the procedure* | | | | | | | | | | | |
| Legibility | | 436 | | 405 | | 31 | | | | | |
| | Yes | | 365 (83.7) | | 345 (94.5) | | 20 5.5) | 3.16 (1.44–6.94) | **0.0041** | 1.07 (0.28–4.15) | 0.92 |

*(Continued)*

**Table 1.** (Continued)

| Variable | Categories | N | All Number (%) | N | Yes Number (%) | N | no Number (%) | Univariate OR (IC95%) | p-value* | Multivariate OR (IC95%) | p-value* |
|---|---|---|---|---|---|---|---|---|---|---|---|
| | | | | | | | | Binary logistic regression | | | |
| | | | | | Participation in the proposal for systematic screening | | | | | | |
| | No | | 71 (16.3) | | 60 (84.5) | | 11 (15.5) | Ref. | Ref | | |
| Accuracy | | 432 | | 401 | | 31 | | | | | |
| | Yes | | 390 (90.3) | | 366 (93.8) | | 24 (6.2) | 3.05 (1.23–7.58) | **0.016** | 0.672 (0.08–5.96) | 0.72 |
| | No | | 42 (9.7) | | 35 (83.3) | | 7 (16.7) | Ref. | Ref | | |
| Praticability | | 432 | | 404 | | 28 | | | | | |
| | Yes | | 359 (83.1) | | 345 (96.1) | | 14 (3.9) | 5.85 (2.65–12.9) | **<0.0001** | 5.64 (1.94–16.4) | **0.0015** |
| | No | | 73 (16.9) | | 59 (80.8) | | 14 (19.2) | Ref | Ref | | |

Intercept: Coefficient±SE (p-value): -0.1256±.1.2502 (p = 0,92)—Ref: reference value for the calculation of l'OR—Shaded cell: variables not retained for the multivariate model—In bold, significant results

*Test performed: logistic regression

extraordinary meetings seems to favour staff participation (p = 0.0044), although this result does not remain significant in the multivariate model.

Finally, in terms of perceived difficulties, only the kit transport (from the nursing home to the collection center stage) seems to be a barrier to participation when difficulties are reported in this stage (p = 0.018). Only one trend can be observed (p = 0.087).

## Discussion

The objective of this study was to understand the factors that increase adoption and the degree of engagement of nursing homes staff to a systematic saliva testing procedure for COVID-19 by direction (or similar function).

The first significant finding was in the adoption rate of the saliva testing programme. Only 8% of the nursing homes contacted (n = 445) chose not to participate to programme, even though the scheme was set up on a spontaneous and gracious basis. This figure demonstrates the extent to which saliva testing could be considered as a possible additional to nasopharyngeal testing, which remains the reference (the official) test in Belgium. The wide acceptance of the system by the nursing homes direction supports the recognition of saliva testing as the new gold standard for the detection of the SARS-CoV-2 virus [22, 23].

The programme participation rate within the nursing homes (49%) was relatively low considering the coverage thresholds necessary for the scheme to be effective. This result reinforces the hypothesis that it is imperative to understand the factors driving staff engagement in a saliva testing scheme.

Based on models that have already described the adoption of preventive health behaviours in a dynamic way [24], we distinguished between factors of initial adoption (by the direction) and those of the level of engagement in the testing system (by the workers themselves). This distinction was particularly relevant given that adoption was a purely direction (or hierarchical) decision and that the level of engagement (participation rate) was directly dependent on the involvement of workers. The identification of different factors favouring adoption and the level of engagement respectively confirmed the need for this distinction and pointed to the

**Table 2. Factors determining the level of staff engagement to a system of systematic screening of saliva testing of staff to control the spread of Covid-19 by Walloon nursing homes (n = 409, December 2020) ($R^2$ = 0.30).**

| Variable | Categories | N | Average participation rate of nursning homes staff during the 3 weeks pilot phase | | Univariate | Regression multiple (n = 340) | |
|---|---|---|---|---|---|---|---|
| | | | Mean | SD | p-value | Coefficient±SE | p-value |
| *Characteristics of the institution* | | | | | | | |
| Sector | | 409 | 0.49 | 0.25 | 0.62 | | |
| | COMM | 204 | 0.51 | 0.26 | | | |
| | ASSOC | 90 | 0.48 | 0.24 | | | |
| | PUBLIC | 115 | 0.48 | 0.25 | | | |
| Geographical Distribution | | 409 | 0.49 | 0.25 | <**0.0001** | | |
| | LIEGE | 121 | 0.40 | 0.25 | | -0.010±.043 | 0.83 |
| | NAMUR | 60 | 0.56 | 0.25 | | .088±.051 | 0.084 |
| | HAINAUT | 160 | 0.54 | 0.23 | | .0803±.043 | 0.061 |
| | LUXEMBOURG | 27 | 0.59 | 0.28 | | 0.14±.063 | **0.030** |
| | BW | 41 | 0.43 | 0.25 | | Ref | Ref |
| Distance from the collection centre | | 408 | 0.50 | 0.25 | 0.91 | | |
| | <11 | 139 | 0.49 | 0.25 | | | |
| | 11–16 | 138 | 0.50 | 0.25 | | | |
| | >16 | 131 | 0.50 | 0.26 | | | |
| Size of nursing homes | | 409 | $R_{Spearman}$ = -0.30 | | <**0.0001** | -0.002±.0003 | <**0.001** |
| *Perceived value of testing* | | | | | | | |
| Justification of testing requirements | | 403 | 0.49 | 0.25 | 0.080 | | |
| | Yes | 349 | 0.50 | 0.25 | | | |
| | No | 54 | 0.44 | 0.26 | | | |
| Screening effectiveness | | 381 | 0.50 | 0.25 | 0.47 | | |
| | Yes | 305 | 0.50 | 0.25 | | | |
| | No | 76 | 0.48 | 0.26 | | | |
| Early detection | | 369 | 0.49 | 0.25 | 0.76 | | |
| | Yes | 302 | 0.50 | 0.25 | | | |
| | No | 67 | 0.48 | 0.27 | | | |
| Perceived added value Compared to Other prevention measures | | 402 | 0.49 | 0.25 | **0.0011** | | |
| | Yes | 358 | 0.51 | 0.25 | | 0.051±.041 | 0.22 |
| | No | 44 | 0.38 | 0.24 | | ref | |
| *Perception of the procedure* | | | | | | | |
| Clarity of screening | | 405 | 0.49 | 0.25 | 0.46 | | |
| | Yes | 345 | 0.50 | 0.26 | | | |
| | No | 60 | 0.47 | 0.24 | | | |
| Clarification of the procedure | | 401 | 0.50 | 0.25 | 0.52 | | |
| | Yes | 366 | 0.50 | 0.25 | | | |
| | No | 35 | 0.47 | 0.25 | | | |
| Praticability | | 404 | 0.49 | 0.25 | 0.64 | | |
| | Yes | 345 | 0.49 | 0.25 | | | |
| | No | 59 | 0.51 | 0.28 | | | |

*(Continued)*

**Table 2.** (Continued)

| Variable | Categories | N | Average participation rate of nursning homes staff during the 3 weeks pilot phase | | Univariate | Regression multiple (n = 340) | |
|---|---|---|---|---|---|---|---|
| | | | Mean | SD | p-value | Coefficient±SE | p-value |
| *Importance of testing* | | | | | | | |
| Personal acceptance of testing by the management | | 408 | 0.50 | 0.25 | **0.0030** | | |
| | Very positive | 325 | 0.52 | 0.25 | | .013±.0.081 | 0.13 |
| | Somewhat positive | 72 | 0.43 | 0.24 | | 0.081±0.085 | 0.34 |
| | Somewhat negative/negative | 11 | 0.32 | 0.27 | | Ref | |
| Priority | | 409 | 0.49 | 0.25 | **<0.0001** | | |
| | High | 368 | 0.51 | 0.25 | | 0.14±0.044 | **0.002** |
| | Medium / weak | 41 | 0.33 | 0.20 | | Ref. | |
| *Decision to participate in testing and support of the staff/management* | | | | | | | |
| Decision making for participation in testing | | 401 | 0.50 | 0.25 | 0.85 | | |
| | Management | 223 | 0.50 | 0.26 | | | |
| | Management Committee | 35 | 0.52 | 0.28 | | | |
| | Employees | 143 | 0.49 | 0.23 | | | |
| Support from management | | 354 | 0.49 | 0.25 | **0.022** | | |
| | Total | 314 | 0.50 | 0.26 | | 0.023±0.052 | 0.66 |
| | Partial | 12 | 0.31 | 0.17 | | -0.11±0.83 | 0.19 |
| | None | 23 | 0.46 | 0.22 | | Ref | |
| Staff support | | 408 | 0.49 | 0.25 | **<0.0001** | | |
| | Yes | 326 | 0.54 | 0.25 | | Ref | |
| | No | 6 | 0.23 | 0.23 | | -0.11±0.83 | 0.19 |
| | Mixed | 76 | 0.32 | 0.20 | | -0.22±.10 | **0.036** |
| *Motivation of staff* | | | | | | | |
| Staff motivation: coordinator | | 409 | 0.49 | 0.25 | 0.86 | | |
| | Not ticked | 379 | 0.50 | 0.25 | | | |
| | ticked | 30 | 0.49 | 0.29 | | | |
| Staff motivation: sent email | | 409 | 0.49 | 0.25 | 0.75 | | |
| | Not ticked | 251 | 0.50 | 0.25 | | | |
| | ticked | 158 | 0.49 | 0.26 | | | |
| Staff motivation: special meeting | | 409 | 0.49 | 0.25 | **0.0044** | | |
| | Not ticked | 273 | 0.47 | 0.26 | | -0.43±0.025 | 0.087 |
| | ticked | 136 | 0.54 | 0.24 | | ref | |
| Staff motivation: tutorial video | | 409 | 0.49 | 0.25 | 0.79 | | |
| | Not ticked | 352 | 0.50 | 0.25 | | | |
| | ticked | 57 | 0.49 | 0.25 | | | |
| Staff motivation:other | | 409 | 0.49 | 0.25 | 0.21 | | |
| | Not ticked | 162 | 0.51 | 0.24 | | | |
| | ticked | 247 | 0.48 | 0.26 | | | |
| Freedom of staff to take part in testing | | 406 | 0.49 | 0.25 | **0.0003** | | |
| | Totally | 263 | 0.46 | 0.26 | | ref | |
| | Incentives | 125 | 0.55 | 0.23 | | 0.0943±0.027 | **< .001** |
| | Imposed | 18 | 0.62 | 0.26 | | 0.10±0.62 | 0.099 |

*(Continued)*

**Table 2.** (Continued)

| Variable | Categories | N | Average participation rate of nursning homes staff during the 3 weeks pilot phase | | Univariate | Regression multiple (n = 340) | |
|---|---|---|---|---|---|---|---|
| | | | Mean | SD | p-value | Coefficient±SE | p-value |
| *Difficulties encountered in the implementation of testing* | | | | | | | |
| Difficulties: transport of the kits | | 408 | 0.49 | 0.25 | **0.018** | | |
| | Yes | 50 | 0.42 | 0.25 | | -0.065±0.038 | 0.087 |
| | No | 358 | 0.51 | 0.25 | | ref | |
| Difficulties: schedule for receiving/depositing kits | | 408 | 0.49 | 0.25 | 0.40 | | |
| | Yes | 84 | 0.47 | 0.25 | | | |
| | No | 324 | 0.50 | 0.25 | | | |
| Difficulties: distribution of the kits | | 408 | 0.49 | 0.25 | 0.44 | | |
| | Yes | 22 | 0.54 | 0.23 | | | |
| | No | 386 | 0.49 | 0.25 | | | |
| Difficulties: self-collection | | 409 | 0.49 | 0.25 | 0.83 | | |
| | Yes | 92 | 0.50 | 0.25 | | | |
| | No | 317 | 0.49 | 0.25 | | | |
| Difficulties: depositing samples in MR/MRS | | 408 | 0.50 | 0.25 | 0.90 | | |
| | Yes | 51 | 0.49 | 0.23 | | | |
| | No | 357 | 0.50 | 0.26 | | | |
| Difficulties: understanding the results | | 408 | 0.50 | 0.25 | 0.55 | | |
| | Yes | 92 | 0.51 | 0.26 | | | |
| | No | 316 | 0.49 | 0.25 | | | |

Ref: reference value

Shaded cell: variables not retained for the multivariate model

need for the promoters of such a system to provide effective and specific support to the facilities that host saliva testing both in the adoption and implementation phases.

In the adoption of saliva testing by nursing homes managers, the perception of the justification of the efforts required by the testing and the perception of the practicality of the procedure were seen to significantly influence the adoption of the system. ORs > 5 indicated the importance of these two variables in the acceptance to the adoption of testing. These two variables were reminiscent of components already identified in Ajzen's [25, 26] theory of planned behaviour under the name of attitude and control over expected behaviour. This attitude is underpinned by beliefs in the expected behaviour. In the present study, it was significant that the cost/benefit balance (approximated by the indicator "Influence of the health situation in the institution on the extent of efforts undertaken for saliva testing") was the most important factor in the beliefs of nursing homes managers. The nursing homes in Belgium have indeed been severely affected (death rate due to COVID) [16], a fact that probably explains the directors views. The practicability of the procedure refers to the control dimension developed by Ajzen in the sense that, in this study, the perceived ability to implement saliva testing was significantly related with the acceptance of screening.

Regarding the level (or intensity) of staff engagement in the saliva testing scheme, workers support and ability to motivate workers' participation were identified **as** major factors in the levels of participation. This result seems obvious given that the success of the scheme depended on their participation. However, this statement can be qualified in that the highest participation rates were observed in the nursing homes where were implemented incentives as well as special meetings to engage staff. These results call for reflection on the development of strategies to promote the use of saliva testing from a health promotion perspective [6]. This approach moves away from a paternalistic policy of imposing preventive measures towards one of supporting autonomy, informed decision-making and empowering individuals in preventive health strategies.

Finally, the results of the study call for questions to be asked about the more technical implementation of screening procedures. In this case, although only showing a trend, the perception of difficulties in transporting the kit proved to be a stumbling block for workers participation. Such difficulties must me further explored to refine the analysis of encountered problems. Difficulties in transport may be related not only to the distance but also to other aspects such as appropriate vehicle for the transport of potential contaminant material. Anyway, the study of difficulties encountered in the implementation of the procedure thus made it possible to identify crucial steps that risk, if left unaddressed, undermining the system.

Within the limits of the study, voluntary participation in the interview-part of the study may create a self-selection bias, leading to an over-representation of nursing homes who took part in the testing scheme. However, the coverage of almost 80% (n = 445) of the population targeted by the survey counterbalances this observation. The management's perception of the scheme is also a weakness that calls for further investigation in order to understand how the direct and indirect beneficiaries of the scheme (workers and residents) experience the situation. The model considered in the study of the level of engagement in testing indicates an $R^2$ of 0.30, which means a moderate explanation of the level of engagement. Another limit must be found is the unique time slot that was dedicated for each nursing home for the collection of salivary samples. It is obviously one of the main reason explaining limited observed participation of staff. It will be necessary to complete this work with other studies, in order to identify the factors linked to the intensity of engagement in such a system in a more systemic way. Finally, it will be necessary to study the engagement of the nursing staff over a longer period of time in order to characterise the factors of maintenance in the prevention system.

## Conclusion

Strategies to prevent the spread of COVID19 involve individual measures in the service of the collective interest. The human and social sciences are of major interest in understanding the adoption of and investment in preventive behaviour. The amount of research available on these human elements remains marginal compared to the more "biomedical" studies of the phenomenon under study. The present study humbly contributes to the understanding of the factors related to a saliva sampling testing device among the workers of the nursing homes in Wallonia. We hope this constitutes a modest contribution to the implementation of effective and efficient prevention programmes in the fight against communicable diseases such as COVID-19.

## Supporting information

**S1 Table. Underlying data set for the study of factors influencing the adoption and participation rate of nursing homes staff in a saliva testing screening programme for COVID-19.** (XLSX)

## Author Contributions

**Conceptualization:** Benoit Pétré, Marine Paridans, Dieudonné Leclercq, Michèle Guillaume.

**Data curation:** Anne-Françoise Donneau, Nadia Dardenne.

**Formal analysis:** Marine Paridans, Anne-Françoise Donneau, Nadia Dardenne.

**Investigation:** Marine Paridans, Nicolas Gillain, Eddy Husson.

**Methodology:** Benoit Pétré, Marine Paridans, Dieudonné Leclercq, Michèle Guillaume.

**Resources:** Christophe Breuer, Fabienne Michel, Margaux Dandoy, Fabrice Bureau, Laurent Gillet.

**Supervision:** Benoit Pétré, Michèle Guillaume.

**Writing – original draft:** Benoit Pétré, Marine Paridans.

**Writing – review & editing:** Benoit Pétré, Marine Paridans, Nicolas Gillain, Eddy Husson, Anne-Françoise Donneau, Nadia Dardenne, Christophe Breuer, Fabienne Michel, Margaux Dandoy, Fabrice Bureau, Laurent Gillet, Dieudonné Leclercq, Michèle Guillaume.

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
