## [Decision Letter · Decision Letter 0]

1 Apr 2022

PONE-D-22-00794Factors influencing the adoption and participation rate of nursing homes staff in a saliva testing screening programme for COVID-19PLOS ONE

Dear Dr. Pétré,

Thank you for submitting your manuscript to PLOS ONE. After careful consideration, we feel that it has merit but does not fully meet PLOS ONE’s publication criteria as it currently stands. Therefore, we invite you to submit a revised version of the manuscript that addresses the points raised during the review process.

We look forward to receiving your revised manuscript.

Kind regards,

Dimitri Beeckman, Ph.D.

Academic Editor

PLOS ONE

Journal Requirements:

2. Please amend your current ethics statement to address the following concerns: Please explain how you recorded/documented participant consent, and if the ethics committees/IRBs approved this consent procedure.

Additional Editor Comments:

Thank you for submitting your work to PLOS ONE. We apologize for the late decision. Given the specificity of the topic, it was not so easy to find reviewers. The reviewers have submitted their reports and the feedback is somewhat conflicting. However, the significance and methodological soundness are sufficiently compelling to warrant further consideration of this manuscript for possible publication. I look forward to receiving the revised version of this important work.

Reviewers' comments:

Reviewer's Responses to Questions

**Comments to the Author**

1. Is the manuscript technically sound, and do the data support the conclusions?

Reviewer #1: Partly

Reviewer #2: Yes

2. Has the statistical analysis been performed appropriately and rigorously? 

Reviewer #1: Yes

Reviewer #2: Yes

3. Have the authors made all data underlying the findings in their manuscript fully available?

Reviewer #1: Yes

Reviewer #2: Yes

4. Is the manuscript presented in an intelligible fashion and written in standard English?

Reviewer #1: Yes

Reviewer #2: Yes

5. Review Comments to the Author

Reviewer #1: Many thanks to the authors for their research in this relevant domain. My specific comments/suggestions can be found in the attached file. In general, the framing in the introduction still needs to be refined, as well as the structuring of the reporting in the Materials and Methods section.

Reviewer #2: This is a report of nation (region) - wide survey of managers and staff of nursing homes in Wallonia (french-speaking part of) Belgium. The study explores the willingness to participate of nursing home managers in a preventive testing programme, using systematic and repeated (during a 3 months period) of saliva-self-tests of staff members during the COVID pandemic. In addition, the real implementation of the programme is explored, using he data on actually performed self-tests.

The study is well powered, almost a full population study, well designed and conducted, and providing usefull information for health policy makers, responsible for pandemic response in this particular sector. The discussion is to the point and well indicating the limitations. The conclusions are within the realm of the findings.

The authors may add more information about the evidence on sensitiviry and speciicity of the selected saliva-seff-test.

6. PLOS authors have the option to publish the peer review history of their article (what does this mean?). If published, this will include your full peer review and any attached files.

Reviewer #1: No

Reviewer #2: **Yes: **Robert H. Vander Stichele, MD, PhD

---

## [Decision Letter · Decision Letter 1]

14 Jun 2022

Factors influencing the adoption and participation rate of nursing homes staff in a saliva testing screening programme for COVID-19

PONE-D-22-00794R1

Dear Dr. Pétré,

We’re pleased to inform you that your manuscript has been judged scientifically suitable for publication and will be formally accepted for publication once it meets all outstanding technical requirements.

Kind regards,

Professor Dimitri Beeckman, Ph.D.

Additional Editor Comments (optional):

Thank you for the adjustments. The reviewers are satisfied with the adjustments made, and so am I. The manuscript has reached the quality for publication. Congratulations.

Reviewers' comments:

Reviewer's Responses to Questions

**Comments to the Author**

1. If the authors have adequately addressed your comments raised in a previous round of review and you feel that this manuscript is now acceptable for publication, you may indicate that here to bypass the “Comments to the Author” section, enter your conflict of interest statement in the “Confidential to Editor” section, and submit your "Accept" recommendation.

Reviewer #1: All comments have been addressed

2. Is the manuscript technically sound, and do the data support the conclusions?

Reviewer #1: Yes

3. Has the statistical analysis been performed appropriately and rigorously? 

Reviewer #1: Yes

4. Have the authors made all data underlying the findings in their manuscript fully available?

Reviewer #1: Yes

5. Is the manuscript presented in an intelligible fashion and written in standard English?

Reviewer #1: Yes

6. Review Comments to the Author

Reviewer #1: Dear authors,

Thank you very much for addressing my previous comments. The changes have significantly improved the manuscript, making it worth publishing in my opinion.

Best regards

7. PLOS authors have the option to publish the peer review history of their article (what does this mean?). If published, this will include your full peer review and any attached files.

Reviewer #1: No

---

## [Editor Report · Acceptance letter]

21 Jun 2022

PONE-D-22-00794R1 

Factors influencing the adoption and participation rate of nursing homes staff in a saliva testing screening programme for COVID-19 

Dear Dr. Pétré:

I'm pleased to inform you that your manuscript has been deemed suitable for publication in PLOS ONE. Congratulations! Your manuscript is now with our production department. 

Kind regards, 

on behalf of

Professor Dimitri Beeckman 

Academic Editor

PLOS ONE